# Sampling CASE Application for the Quality Control of Published Natural Product Structures

**DOI:** 10.3390/molecules26247543

**Published:** 2021-12-13

**Authors:** Lorena Martins Guimarães Moreira, Jochen Junker

**Affiliations:** Oswaldo Cruz Foundation—CDTS, Rio de Janeiro 21040-900, Brazil; mglorena21@gmail.com

**Keywords:** theoretical NMR data, CASE, natural products, web-tool

## Abstract

Structure elucidation with NMR correlation data is dicey, as there is no way to tell how ambiguous the data set is and how reliably it will define a constitution. Many different software tools for computer assisted structure elucidation (CASE) have become available over the past decades, all of which could ensure a better quality of the elucidation process, but their use is still not common. Since 2011, WebCocon has integrated the possibility to generate theoretical NMR correlation data, starting from an existing structural proposal, allowing this theoretical data then to be used for CASE. Now, WebCocon can also read the recently presented NMReDATA format, allowing for uncomplicated access to CASE with experimental data. With these capabilities, WebCocon presents itself as an easily accessible Web-Tool for the quality control of proposed new natural products. Results of this application to several molecules from literature are shown and demonstrate how CASE can contribute to improve the reliability of Structure elucidation with NMR correlation data.

## 1. Introduction

Software tools for computer assisted structure elucidation (CASE) of small molecules have been under development for over 50 years now. The methods used can roughly be divided into stochastic (S), deterministic (D), and hybrid (H), with hybrid representing various different combinations of stochastic and deterministic methods. The most prominent methods are fragment assemblers (H) [1,2,3,4,5,6], expert systems (H) [7,8,9], databases of ${}^{13}$C NMR chemical shifts or neural networks (S) [10,11,12,13], structure generation by reduction (S/H) [14], logic engines (D) [15], stochastic structure generators (S) [16], combinatorial brute force (D) [17,18,19,20,21], combinatorial brute force with restraints (D/H) [22,23], genetic algorithms (S) [24,25,26], simulated annealing (S) [27], convergent structure generation (S) [28,29], evolutionary algorithm (S) [30], fuzzy structure generation (S) [31], and expert systems with DFT (H) [32]. Altogether, 14 different methods were implemented in more than 20 different tools, some of which are freely usable, and many of them have not seen further development by their authors after the first publication. It can be observed that, out of the eight stochastic methods, only database methods are still actively developed, whereas both purely deterministic methods see ongoing development. However, the most active development happens for some of the hybrid methods, that combine stochastic and deterministic elements in various ways. Despite of all these efforts, none of these tools has become popular enough to be used for quality control of published structures by a wider community of researchers or journal editors. The ${}^{13}$C chemical shift-based tools have an outstanding position in this list, as they do not take NMR correlation data into account. They can either map chemical shifts to chemical neighborhoods, which in turn are assembled into a proposed constitution, or determine chemical neighborhoods for each carbon of a proposed constitution and predict a chemical shift. One such free to use tool is CSEARCH [10,13,33], which allows for an automated verification of ${}^{13}$C chemical shift assignments for a proposed structure combining a database and a neural network verification. This approach might fail if few similar structures are available, or if incorrect data is contained in the database. For such applications, tools using NMR correlation data are better suited. One tool that uses NMR correlation data is Cocon with the online interface WebCocon, which has been improved continuously over the past 20 years [22,34,35,36,37,38,39]. Today, Cocon can be used freely via a web interface called WebCocon, that facilitates the input file creation, submission, and visualization of the results.

In a first step towards the use of CASE for quality control, the creation of theoretical NMR correlations from an existing constitution was implemented in WebCocon [37]. This way, the suggested structure is drawn on the web page, theoretical NMR correlations are created, and then they are submitted to a CASE run. It might seem simplistic, but, if WebCocon produces more than one structural proposal under these circumstances, it means that the constitution cannot be verified using NMR correlation data alone, and the use of other information is needed. One possibility is the inclusion of NOEs, which was explored for the first time in combination with genetic algorithms [26]. However, this approach was abandoned due to the very complex parametrization of the genetic algorithm, that defied automation [40]. Since then, NOEs have been used only after the constitution has been defined, in order to determine the stereochemistry of the resulting molecule [41,42]. WebCocon is the first CASE tool that can use NOEs in the ranking of the structural proposals, in a similar way as NOE data is frequently used in publications, but now backed up by structure calculations. It was already shown that, even with coarsely defined NOEs, the conformations obtained as results were similar to published X-ray structures. [39]. The exploration of ${}^{13}$C NMR chemical shift calculation is commonly accepted for the comparison of structural proposals, and it is carried out manually here. This is a first step towards a future quality control method for the structure discussion used in publications.

With the popularization of file formats, such as NMReDATA [43,44,45], the structure verification could be carried out automatically using experimental data. WebCocon can read NMReDATA files, thus offering the possibility of a structure discussion based on theoretical and experimental data.

New natural products published over the past years, whose structures were elucidated using NMR data were chosen from the literature and submitted to a structure discussion using WebCocon. As the molecules generally were not available as electronic files, structures were drawn in WebCocon. Theoretical NMR correlation data for ${}^{3}J$_HH_-COSY and ${}^{2,3}J$_HC_-HMBC correlations (${}^{2,3}J$_HN_-HMBC and 1,1-ADEQUATE correlations are generated but were not used) with carbon chemical shifts from an internal lookup table were created automatically for the structures, and the data was submitted to a CASE run. If more than one structure is proposed, the ranked results are compared automatically to the published structure. Additionally, the suggested alternative structures were compared manually to the experimental structures by back-calculating ${}^{13}$C chemical shifts using CS Chemdraw V15. When chemical shifts were used for the discussion, only the atoms with the largest expected deviations were selected, and the comparison was based on the average deviation observed ($\bar{\Delta }$). Looking for further automation possibilities, the use of CSEARCH as an alternative to CS Chemdraw was investigated for one example, and, for two molecules, the use of the standard deviation (σ) of the chemical shift deviations and their range was included. A standard deviation of 5.5 ppm is expected for the values back-calculated in CS Chemdraw, according to the manual.

${}^{3}J$_HH_-COSY and ${}^{2,3}J$_HC_-HMBC correlations are the most common experiments acquired that contain information about the heavy atom skeleton of a molecule. The three-bond-long COSY correlation between two hydrogen atoms indicates that the two bearing heavy atoms are directly connected. As both heavy atoms are required to be bonded to hydrogens, this experiment provides only limited information for proton poor compounds. The ${}^{2,3}J$_HC_-HMBC correlations are observed between a hydrogen and a carbon that are two or three bonds away. This experiment requires only one hydrogen for the observation of the correlation; thus, it can be used to localize carbons without protons in the skeleton. However, as the observed correlations can originate over two or three bonds, and differentiation normally is not possible, the incorporation of this data is more difficult. The use of CASE for the interpretation is recommended for this ambiguous data. Other experiments have been developed over the years to provide additional information about a molecule’s skeleton, most prominently the ${}^{2,3}J$_HN_-HMBC, 1,1-ADEQUATE, and H2BC. The ${}^{2,3}J$_HN_-HMBC is similar to the carbon version described before but correlates to a nitrogen. The 1,1-ADEQUATE [22,34,35,46,47] and H2BC [48,49,50] are experiments that provide a ${}^{2}J$_HC_-HMBC type of correlation and can be used complimentary to the aforementioned ${}^{2,3}J$_HC_-HMBC data. Theoretical ${}^{2,3}J$_HN_-HMBC and 1,1-ADEQUATE correlations were not used in this analysis because this data is not routinely acquired and, therefore, also not included into publications.

WebCocon is a web-based interface to the program Cocon. Initially, WebCocon helps in the creation of the input file for Cocon, offering different functionalities for this purpose. With this input file, Cocon creates a list of all constitutions that are compatible with the provided correlation data. These constitutions are ranked on the server by different methods and then transferred back to the web interface for visualization. Alternatively, the suggested constitutions can be downloaded in a SDF file and inspected offline by the user.

## 2. Results

The results resumed in Table 1 are just samples from the literature. Further automation in the ranking process will streamline the analysis for many more molecules.

### 2.1. Luteolin 8-C-E-Propenoic Acid

Luteolin 8-C-E-propenoic acid [51] was published as molecule **1-A** from Figure 1. WebCocon suggested 25 constitutions with the theoretical NMR correlation data set, including the published molecule. Out of those, only 6 underwent further inspection, as the others contained very restrained ring systems. The force field total energies calculated strongly favor molecules **1-A** and **1-B**.

The distinction of **1-A** from the other suggested constitutions can be done using ${}^{13}$C chemical shifts of selected atoms, when comparing experimental data with back-calculated data in Table 2. The back-calculated ${}^{13}$C chemical shift values for all carbons of all verified constitutions are in Figure A1.

### 2.2. Tomentodiplacone

Tomentodiplacone [52] was published as molecule **2-A** from Figure 2. WebCocon suggested 2 constitutions with the theoretical NMR correlation data set, including the published molecule. The force field total energy calculated for both molecules is almost equal.

The distinction of **2-A** from the other suggested constitution can be done using ${}^{13}$C chemical shifts of selected atoms. The average deviation of the back-calculated chemical shifts from the experimental values is considerably smaller for **2-A**, as shown in Table 3. The back-calculated ${}^{13}$C chemical shift values for all carbons of all verified constitutions are in Figure A2.

### 2.3. Kadangustin A

Kadangustin A [53] was published as molecule **3-A** from Figure 3. WebCocon suggested 13 constitutions with the theoretical NMR correlation data set, 5 of which are shown here, including the published molecule. The force field total energy calculated for these 5 molecules is very similar.

The distinction of **3-A** from the other suggested constitutions is possible using ${}^{13}$C chemical shifts of some selected atoms, when comparing experimental data with back-calculated data in Table 4 using the average deviation ($\bar{\Delta }$). However, the standard deviation (σ) and range (ΔmaxΔmin) clearly favor **3-D** as solution, in spite of the higher total energy. This case should be verified in more detail. The back-calculated ${}^{13}$C chemical shift values for all carbons of all verified constitutions are in Figure A3.

### 2.4. Berkeleyamide D

Berkeleyamide D [54] was published as molecule **4-A** from Figure 4. WebCocon suggested 2 constitutions with the theoretical NMR correlation data set, including the published molecule. The force field total energy calculated for both molecules clearly favors **4-B**.

The confirmation of **4-A** as correct solution is also not possible using ${}^{13}$C and ${}^{1}$H chemical shift values of selected atoms, as the average deviation is smaller for **4-B**, as shown in Table 5. The back-calculated ${}^{13}$C and ${}^{1}$H chemical shift values using CS Chemdraw for all atoms of both verified constitutions are in Figure A4 and Figure A5, respectively.

Although the average deviations shown in Table 5 indicate that **4-B** matches the experimental chemical shift values better, for both suggested constitutions, there is one carbon with a chemical shift deviation ≫ 10 ppm (carbon 8 for **4-A** and carbon 11 for **4-B**). Therefore, all ${}^{13}$C chemical shift values back-calculated by CS Chemdraw (**M-I**) and CSEARCH (**M-II**) for all carbon atoms were included in a comparison in Table 6.

The standard deviation σ and range ΔmaxΔmin observed in the comparison of the experimental and the different back-calculated chemical shift values using both methods lead to a contradictory result. Whereas method **M-I** favors **4-B**, method **M-II** favors **4-A** as correct constitution. However, for both cases, the range ΔmaxΔmin is very high, indicating that neither one of the constitutions might be correct. An analysis with experimental correlation data would be be the next step, followed by a new verification of back-calculated carbon chemical shift values for all alternatives. If the NMR data was available as NMReDATA, this would be straightforward.

### 2.5. 14-Norpseurotin A

14-norpseurotin A [55] was published as molecule **5-A** from Figure 5. WebCocon suggested 13 constitutions with the theoretical NMR correlation data set, including the published molecule. The force field total energies calculated favors **5-B** over **5-A**; all other alternatives show considerably higher values.

The distinction of **5-A** from **5-B** is also not possible using ${}^{13}$C chemical shifts, when comparing experimental data with back-calculated data for selected atoms, as can be seen in Table 7. Actually, for **5-B** the back-calculated ${}^{13}$C and ${}^{1}$H chemical shifts match the experimental values better, as can be verified by the average deviation. This case should be verified in more detail. The back-calculated ${}^{13}$C and ${}^{1}$H chemical shift values for all verified constitutions are in Figure A6 and Figure A7, respectively.

### 2.6. Feruloylpodospermic Acid A

Feruloylpodospermic acid A [56] was published as molecule **6-A** from Figure 6. WebCocon suggested 2 constitutions with the theoretical NMR correlation data set, including the published molecule. The force field total energy calculated for both molecules slightly favors **6-B**.

A comparison of the experimental and back-calculated ${}^{13}$C chemical shifts for the 5 carbons numbered in Figure 6 is shown in Table 8. The observed average deviation also slightly favors **6-B**. This case should be verified in more detail. The back-calculated ${}^{13}$C chemical shift values for all carbons of all verified constitutions are in Figure A8.

### 2.7. Cochinchistemoninone

Cochinchistemoninone [57] was published as molecule **7-A** from Figure 7. WebCocon suggested 3 constitutions with the theoretical NMR correlation data set, including the published molecule. The force field total energy calculated for the molecules favors **7-C**.

A comparison of the experimental and back-calculated ${}^{13}$C chemical shifts for the 5 carbons numbered in Figure 7 is shown in Table 9. The observed average deviation favors **7-A** as correct solution, but, in all suggested constitutions, the experimental value of the chemical shift of carbon 2 is considerably different from the back-calculated value. This case should be verified in more detail. The back-calculated ${}^{13}$C chemical shift values for all carbons of all verified constitutions are in Figure A9.

### 2.8. Milicifoline B

Milicifoline B [58] was published as molecule **8-A** from Figure 8. When using theoretical NMR correlation data from **8-A**, WebCocon suggested 3 constitutions, including the published molecule. The force field total energy calculated for the three molecules is very similar.

The experimental and back-calculated ${}^{13}$C chemical shifts for two carbons of all constitutions are shown in Table 10. Due to the large average deviation, the suggested constitution **8-C** can be excluded as possible solution. The average deviation favors **8-B** over **8-A**, but without a strong differentiation. The back-calculated ${}^{13}$C chemical shift values for all carbons of all verified constitutions are in Figure A10.

In their publication, the authors used 3 NOEs (see Figure 9) in order to decide between the two possible constitutions, so this information was also included into WebCocon.

The distances measured after the WebCocon run are shown in Table 11, and they favor the originally published constitution. However, the distances measured in the calculated molecules are all > 500 pm and, therefore, at the limit of measurable NOEs [59]. Due to this, no reliable decision between the two suggested constitutions can be made.

### 2.9. 5α-Cyprinol Sulfate

5α-cyprinol sulfate [60] was published as molecule **9-A** from Figure 10. When using theoretical NMR correlation data from **9-A**, WebCocon suggested only one solution. However, for this molecule, experimental data is available in the NMReDATA format. Thus, the WebCocon analysis was repeated using the published experimental data, resulting in 4 suggested constitutions, including the published molecule. The force field total energy calculated for the four molecules is very similar.

All constitutions have the same skeleton; they differ only in the positioning of the sulfate group. Hence, ${}^{13}$C chemical shifts are expected to vary mainly for the carbons adjacent to the sulfate group. The experimental and back-calculated ${}^{13}$C chemical shifts for these 4 carbons of all constitutions are shown in Table 12. The calculated average deviation over the 4 carbons for the suggested constitutions clearly favors the constitution **9-A**, matching the publication. The back-calculated ${}^{13}$C chemical shift values for all carbons of all verified constitutions are in Figure A11.

### 2.10. Viridiol aka TAEMC161

Viridiol (**10-C**) was originally published in 1969, as derivative of viridin [61]. The crystal structure was published in 2013 [62], and, in recent years, the total synthesis has become available [63,64,65,66]. In 2000, TAEMC161 was isolated as metabolite from *Trichoderma hamatum*, and **10-B** was suggested as structure [67]. Shortly thereafter, the structure was revised as being identical to viridiol [68]. In 2010, the structure revision was reviewed by the use of CASE software, that confirmed previous findings [69]. The revisions were based mainly on the comparison of experimental and calculated carbon chemical shifts, using DFT and database methods.

The theoretical correlation data-based analysis of TAEMC161 reveals that this constitution could be described unambiguously by NMR correlation data. The same analysis with viridiol results in the 3 constitutions with similar total energies shown in Figure 11 and three more results with complex bridged ring systems that are not shown, with much higher total energies. From these 3 constitutions, **10-B** is the originally published metabolite TAEMC161, meaning that NMR correlation data cannot distinguish between viridiol and TAEMC161. Thus, other means of distinction need to be used. For this analysis, we have performed carbon chemical shift calculation for all atoms in all 3 constitutions using several methods: **M-I** was CSEARCH, **M-II** was CS Chemdraw, **M-IIIa** DFT with ORCA v5.0.1 using the conformation obtained by WebCocon and **M-IIIb** DFT with ORCA v5.0.1 using a DFT optimized conformation. The results are shown in Table 13 for **10-A**, Table 14 for TAEMC161 (**10-B**), and Table 15 for viridiol (**10-C**).

The resulting RMSD values for the chemical shift deviations observed for the different methods and constitutions show that the empirical methods outperform the DFT method in all conditions, as has already been observed previously [70,71]. In this case, **10-B** and **10-C** cannot be distinguished by DFT, whereas both empirical methods clearly favor **10-C** as solution. Thus, the analysis confirms the published revisions but also proves that correlations data of viridiol are alone are not enough for a distinction of **10-B** and **10-C**.

## 3. Discussion & Conclusions

CASE revealed more than one solution for all molecules discussed here. Since the suggested constitutions are compatible with the theoretical NMR correlation data set of the originally published structure, other means of identification of the correct constitution had to be explored. First, an inspection with “a chemists eye” of the results was carried out, supported by force field total energies, as WebCocon might have generated structures that are not likely to exist. In one case, the inclusion of NOE data from the original publication favors one of the possible solutions, but was not enough for a decision. Secondly, calculation of ${}^{13}$C NMR chemical shifts for the suggested constitutions was carried out and compared to the experimental data. However, for most examples, this was also not enough for a final decision.

The obtained results suggest that many of the new structures published using NMR correlation data could not be described unambiguously by theoretical NMR correlation data. Taking into account that the experimental data sets usually contain fewer data, even more new structures could be subjected to questions. Surveys of revised structures [32,69,71,72,73] clearly show that improvements as to how new small molecules are published are urgently needed. The use any of the software tools for CASE available, together with the popularization of more comprehensive data formats, such as NMReDATA, will lead to an improved traceability of the structure elucidation process, as shown here for 5α-cyprinol sulfate. As it is, WebCocon already could easily be integrated into a quality control workflow.

## Figures and Tables

**Figure 1 molecules-26-07543-f001:**
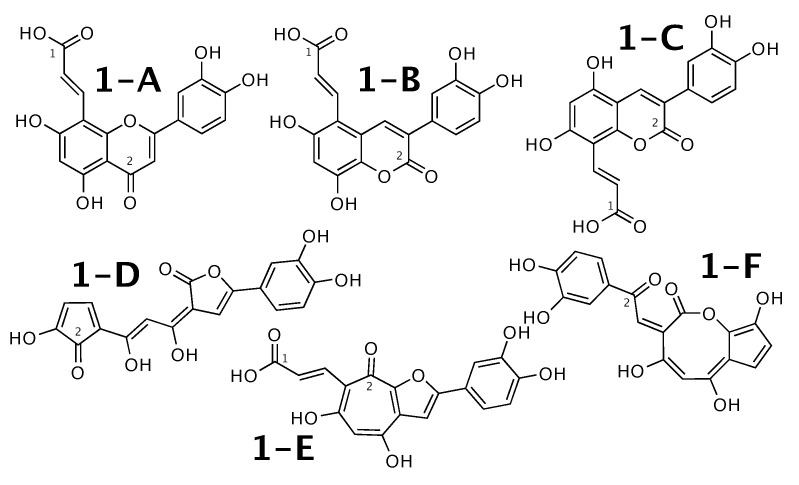
Luteolin 8-C-E-propenoic acid (**1-A**) and the 5 alternative constitutions suggested by WebCocon.

**Figure 2 molecules-26-07543-f002:**
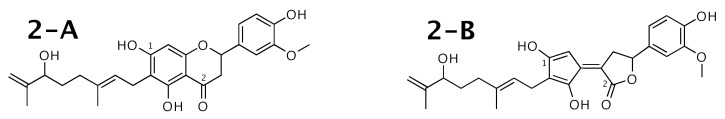
Tomentodiplacone (**2-A**) and the alternative constitution suggested by WebCocon. ${}^{13}$C chemical shifts were back-calculated for the atoms numbered in the molecules.

**Figure 3 molecules-26-07543-f003:**
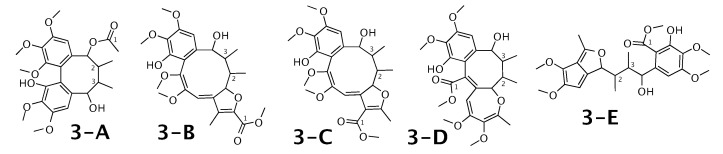
Kadangustin A (**3-A**) and the 4 alternative constitutions suggested by WebCocon. ${}^{13}$C chemical shifts were back-calculated for the atoms numbered in the molecules.

**Figure 4 molecules-26-07543-f004:**
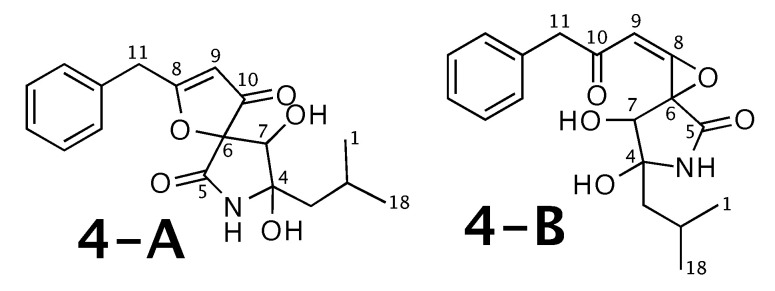
Berkeleyamide D (**4-A**) and the alternative constitution suggested by WebCocon. ${}^{13}$C and ${}^{1}$H chemical shifts were back-calculated for the atoms numbered in the molecules.

**Figure 5 molecules-26-07543-f005:**
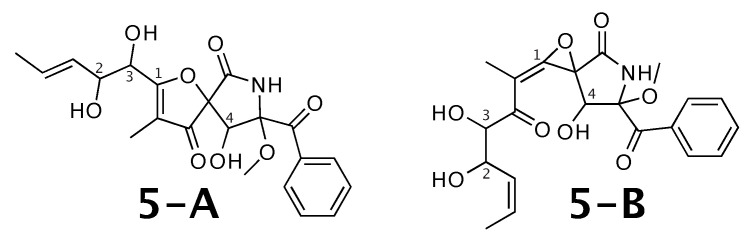
14-norpseurotin A (**5-A**) and the alternative constitution suggested by WebCocon. ${}^{13}$C and ${}^{1}$H chemical shifts were back-calculated for the atoms numbered in the molecules.

**Figure 6 molecules-26-07543-f006:**
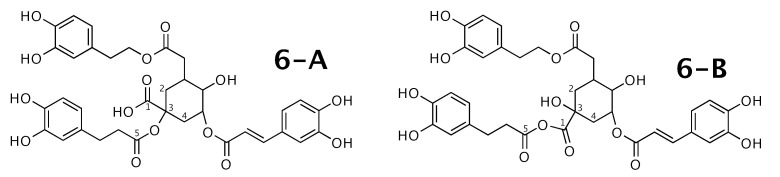
Feruloylpodospermic acid A (**6-A**) and the alternative constitution suggested by WebCocon. ${}^{13}$C chemical shifts were back-calculated for the atoms numbered in the molecules.

**Figure 7 molecules-26-07543-f007:**
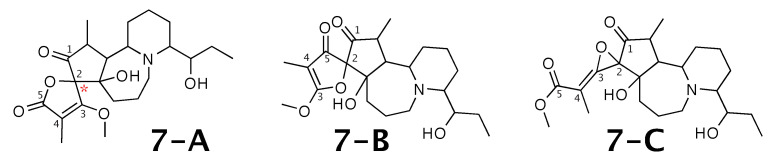
Cochinchistemoninone (**7-A**) and the alternative constitution suggested by WebCocon. ${}^{13}$C chemical shifts were back-calculated for the atoms numbered in the molecules.

**Figure 8 molecules-26-07543-f008:**
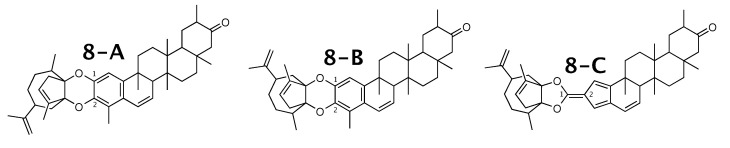
Milicifoline B (**8-A**) and the 2 alternative constitution suggested by WebCocon.

**Figure 9 molecules-26-07543-f009:**
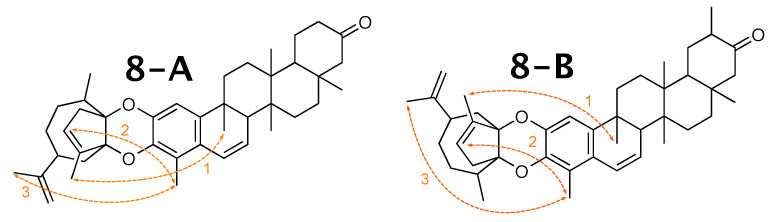
NOEs (depicted in orange) used to identify the correct conformation of milicifoline B (**8-A**) in the literature and the corresponding contacts in **8-B**.

**Figure 10 molecules-26-07543-f010:**
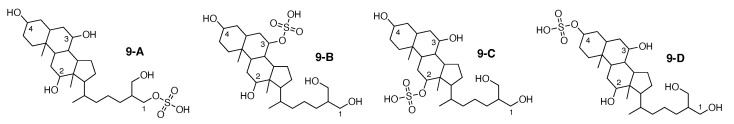
5α-cyprinol sulfate (**9-A**) and the 3 alternative constitutions suggested by WebCocon.

**Figure 11 molecules-26-07543-f011:**
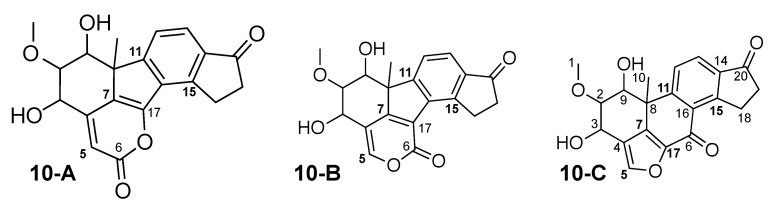
Viridiol (**10-C**), original TAEMC161 (**10-B**) and one other alternative constitution, all suggested by WebCocon. The positions with largest carbon chemical shift variations are marked in bold.

**Table 1 molecules-26-07543-t001:** Overview of the results.

Molecule	# Inspected	Confirmed
Luteolin 8-C-E-propenoic acid	6	yes
Tomentodiplacone	2	yes
Kadangustin A	5	no
Berkeleyamide D	2	no
14-norpseurotin A	2	no
Feruloylpodospermic acid A	2	no
Cochinchistemoninone	3	no
Milicifoline B	3	pending further analysis
5α-cyprinol sulfate	4	yes
viridiol aka TAEMC161	>3	yes

**Table 2 molecules-26-07543-t002:** Comparison of experimental and back-calculated ${}^{13}$C chemical shifts for luteolin 8-C-E-propenoic acid (**1-A**) and the 5 suggested alternative constitutions. ${}^{13}$C chemical shifts were back-calculated for the atoms numbered in the molecules. $\bar{\Delta }$ is the average of the deviations of the calculated to the experimental values. E_total_ is the total force field energy calculated in kcal/mol for the constitutions.

	$\delta^{13}$C [ppm]						
**Atom**	**Exp.**	**1-A**	**1-B**	**1-C**	**1-D**	**1-E**	**1-F**
1	169.0	171.5	171.5	171.5	-	170.4	-
2	182.2	182.1	160.5	160.5	187.0	175.0	189.7
$\bar{\Delta }$		1.3	12.1	12.1	2.4	4.3	3.8
E_total_ [kcal/mol]		26	29	45	80	74	303

**Table 3 molecules-26-07543-t003:** Comparison of experimental and back-calculated ${}^{13}$C chemical shifts for tomentodiplacone (**2-A**) and the suggested alternative constitution. $\bar{\Delta }$ is the average of the deviations of the calculated to the experimental values. E_total_ is the total force field energy calculated in kcal/mol for the constitutions.

	$\delta^{13}$C [ppm]		
**Atom**	**Exp.**	**2-A**	**2-B**
1	164.4	162.5	182.6
2	196.4	196.8	170.0
$\bar{\Delta }$		1.2	22.3
E_total_ [kcal/mol]		62	64

**Table 4 molecules-26-07543-t004:** Comparison of experimental and back-calculated ${}^{13}$C chemical shifts for kadangustin A (**3-A**) and the 4 suggested alternative constitutions. $\bar{\Delta }$ is the average deviation, σ is the standard deviation of the chemical shifts, and ΔmaxΔmin is the range of the deviations of the calculated chemical shifts to the experimental values. E_total_ is the total force field energy calculated in kcal/mol for the constitutions.

	$\delta^{13}$C [ppm]					
**Atom**	**Exp.**	**3-A**	**3-B**	**3-C**	**3-D**	**3-E**
1	172.0	170.2	162.1	165.0	165.4	171.4
2	39.3	31.3	33.6	33.6	35.9	46.7
3	34.8	34.8	38.7	38.7	31.1	41.8
$\bar{\Delta }$		3.3	6.5	5.5	4.6	5.0
σ		4.2	7.1	6.0	1.8	4.5
ΔmaxΔmin		8.0	13.8	10.9	6.6	8.0
E_total_ [kcal/mol]		160	165	160	168	156

**Table 5 molecules-26-07543-t005:** Comparison of experimental and back-calculated ${}^{13}$C and ${}^{1}$H chemical shifts for berkeleyamide D (**4-A**) and the suggested alternative constitution. $\bar{\Delta }$ is the average of the deviations of the calculated to the experimental values. E_total_ is the total force field energy calculated in kcal/mol for the constitutions. Deviations ≫ 10 ppm are labeled with “*”.

	$\delta^{13}$C [ppm]			$\delta^{1}$H [ppm]		
**Atom**	**Exp.**	**4-A**	**4-B**	**Exp.**	**4-A**	**4-B**
4	84.9	76.6	87.2			
6	95.3	100.1	90.2			
7				4.41	4.42	4.0
8	197.7	165.5 *	198.0			
9	104.4	105.6	98.8	5.35	4.85	5.1
10	199.4	204.9	196.5			
11	37.4	45.0	55.7 *	3.98	3.22	4.2
$\bar{\Delta }$		8.5	4.9		0.5	0.3
E_total_ [kcal/mol]		67	24			

**Table 6 molecules-26-07543-t006:** Comparison of experimental and back-calculated ${}^{13}$C chemical shifts for all carbons in berkeleyamide D (**4-A**) and the suggested alternative constitution, using CS Chemdraw (**M-I**) and CSEARCH (**M-II**). σ is the standard deviation and ΔmaxΔmin is the range of the deviations between the back-calculated and experimental value.

		4-A		4-B	
**Atom**	**Exp.**	**M-I**	**M-II**	**M-I**	**M-II**
1	23.9	23.5	23.8	23.5	23.8
2	24.0	23.6	24.0	23.9	24.0
3	45.5	47.1	45.5	44.3	45.4
4	84.9	76.6	84.9	87.2	85.5
5	164.1	169.2	171.6	169.2	170.2
6	95.3	100.1	95.3	90.2	65.4
7	75.1	76.5	75.1	78.5	75.7
8	197.8	204.9	181.2	198.0	156.9
9	104.4	105.6	104.4	98.8	89.6
10	199.4	165.5	199.4	196.5	198.4
11	37.4	45.0	43.0	55.7	47.5
12	133.2	137.4	133.2	135.6	134.6
13	129.2	129.0	129.2	129.6	129.6
14	129.2	128.6	129.2	129.2	128.9
15	127.0	125.7	127.1	127.6	127.1
16	129.2	128.6	129.2	129.2	128.9
17	129.2	129.0	129.2	129.6	129.6
18	23.8	23.5	23.8	23.5	23.8
σ		9.0	4.6	5.1	12.5
ΔmaxΔmin		41.5	24.1	23.9	51.0

**Table 7 molecules-26-07543-t007:** Comparison of experimental and back-calculated ${}^{13}$C and ${}^{1}$H chemical shifts for 14-norpseurotin A (**5-A**) and the suggested alternative constitution. $\bar{\Delta }$ is the average of the deviations of the calculated to the experimental values. E_total_ is the total force field energy calculated in kcal/mol for the constitutions.

	$\delta^{13}$C [ppm]			$\delta^{1}$H [ppm]		
**Atom**	**Exp.**	**5-A**	**5-B**	**Exp.**	**5-A**	**5-B**
1	185.7	167.3	188.0			
2	70.4	69.7	69.8	4.78	4.0	4.2
3	70.3	71.6	88.9	4.62	4.0	4.2
4	73.2	68.1	70.1	4.70	5.0	4.8
$\bar{\Delta }$		6.4	6.2		0.6	0.4
E_total_ [kcal/mol]		160	148			

**Table 8 molecules-26-07543-t008:** Comparison of experimental and back-calculated ${}^{13}$C chemical shifts for feruloylpodospermic acid A (**6-A**) and the suggested alternative constitution. $\bar{\Delta }$ is the average of the deviations of the calculated to the experimental values. E_total_ is the total force field energy calculated in kcal/mol for the constitutions.

	$\delta^{13}$C [ppm]		
**Atom**	**Exp.**	**6-A**	**6-B**
1	170.4	171.0	173.2
2	38.9	39.2	35.9
3	83.5	78.9	85.3
4	34.1	38.4	34.0
5	173.1	170.2	173.1
$\bar{\Delta }$		2.5	1.5
E_total_ [kcal/mol]		75	56

**Table 9 molecules-26-07543-t009:** Comparison of experimental and back-calculated ${}^{13}$C chemical shifts for cochinchistemoninone (**7-A**) and the 2 suggested alternative constitutions. $\bar{\Delta }$ is the average of the deviations of the calculated to the experimental values. E_total_ is the total force field energy calculated in kcal/mol for the constitutions.

	$\delta^{13}$C [ppm]			
**Atom**	**Exp.**	**7-A**	**7-B**	**7-C**
1	208.9	211.2	211.3	211.3
2	90.6	117.6	127.4	103.3
3	169.2	171.8	182.6	191.4
4	98.4	98.7	88.1	95.6
5	173.1	175.2	197.9	167.3
$\bar{\Delta }$		6.9	17.5	9.2
E_total_ [kcal/mol]		116	87	80

**Table 10 molecules-26-07543-t010:** Comparison of experimental and back-calculated ${}^{13}$C chemical shifts for milicifoline B (**8-A**) and the 2 suggested alternative constitutions. $\bar{\Delta }$ is the average of the deviations of the calculated to the experimental values. E_total_ is the total force field energy calculated in kcal/mol for the constitutions.

	$\delta^{13}$C [ppm]			
**Atom**	**Exp.**	**8-A**	**8-B**	**8-C**
1	144.1	145.5	145.5	139.6
2	142.8	136.9	144.4	91.7
$\bar{\Delta }$		3.6	1.5	27.8
E_total_ [kcal/mol]		267	288	265

**Table 11 molecules-26-07543-t011:** Comparison of back-calculated distances for the NOEs for milicifoline B (**8-A**) and the suggested alternative constitution **8-B**.

	Δs [pm]	
**NOE**	**8-A**	**8-B**
1	705	850
2	612	618
2	503	832

**Table 12 molecules-26-07543-t012:** Comparison of experimental and back-calculated ${}^{13}$C chemical shifts for 5α-cyprinol sulfate (**9-A**) and the 3 suggested alternative constitutions for selected carbons. $\bar{\Delta }$ is the average of the deviations of the calculated to the experimental values. E_total_ is the total force field energy calculated in kcal/mol for the constitutions.

	$\delta^{13}$C [ppm]				
**Atom**	**Exp.**	**9-A**	**9-B**	**9-C**	**9-D**
1	67.6	77.7	64.2	64.2	64.2
2	72.3	73.3	73.3	73.3	87.6
3	68.4	68.4	91.5	68.4	68.4
4	71.4	71.4	71.4	89.5	71.4
$\bar{\Delta }$		2.8	6.9	5.6	4.7
E_total_ [kcal/mol]		42	48	56	53

**Table 13 molecules-26-07543-t013:** Comparison of experimental and back-calculated ${}^{13}$C chemical shifts for all carbons in **10-A**. The largest deviations are bold in the table.

Atom	Exp.	M-I	M-II	M-IIIa	M-IIIb
1	60.7	60.8	57.8	61.0	60.4
2	81.7	82.0	88.9	93.6	89.5
3	61.6	70.4	66.1	68.8	69.3
4	122.1	138.8	164.0	154.2	158.8
**5**	**145.6**	**116.1**	**110.4**	**126.6**	**111.6**
6	173.4	161.1	162.6	174.0	165.8
**7**	**142.4**	**118.6**	**116.5**	**133.2**	**130.4**
8	42.3	47.3	44.8	55.5	55.1
9	71.7	71.9	80.7	80.0	75.9
10	30.5	28.0	21.0	22.7	22.1
**11**	**158.6**	157.2	**145.4**	**172.3**	**169.7**
12	127.3	125.4	125.1	131.4	128.8
13	127.3	126.6	123.7	131.6	130.9
14	129.8	133.9	133.9	142.8	144.9
**15**	158.0	150.7	137.8	154.4	158.5
16	136.9	131.5	124.5	136.9	137.6
**17**	**145.7**	146.4	148.6	**167.9**	**172.1**
18	28.4	29.0	24.3	28.8	28.6
19	36.5	35.7	36.6	38.3	39.3
20	206.7	206.8	206.2	213.0	210.5
RMSD		10.2	15.6	12.1	14.4

**Table 14 molecules-26-07543-t014:** Comparison of experimental and back-calculated ${}^{13}$C chemical shifts for all carbons in TAEMC161 (**10-B**). The largest deviations are bold in the table.

Atom	Exp.	M-I	M-II	M-IIIa	M-IIIb
1	60.7	60.8	57.8	61.6	59.8
2	81.7	81.7	88.9	99.4	96.9
3	61.6	69.1	62.9	79.2	76.1
4	122.1	122.1	127.5	129.0	123.8
**5**	145.6	138.3	149.3	**158.5**	**158.4**
6	173.4	158.0	153.4	171.1	163.0
**7**	142.4	142.4	**153.0**	**165.6**	**174.0**
8	42.3	53.7	48.0	58.6	59.1
9	71.7	71.1	80.7	83.2	83.3
10	30.5	20.5	21.0	21.1	20.0
**11**	158.6	158.6	**145.4**	**171.9**	**166.3**
12	127.3	122.0	125.1	131.4	129.1
13	127.3	127.3	123.7	131.0	129.0
14	129.8	136.9	133.9	143.3	145.4
**15**	158.0	**139.2**	**137.8**	157.8	159.8
16	136.9	129.6	134.0	140.2	141.9
**17**	145.7	**123.5**	**124.5**	**138.0**	**129.5**
18	28.4	28.4	24.1	32.9	31.3
19	36.5	36.5	36.6	39.0	39.5
20	206.7	206.7	206.2	214.2	211.6
RMSD		8.8	9.8	11.0	12.0

**Table 15 molecules-26-07543-t015:** Comparison of experimental and back-calculated ${}^{13}$C chemical shifts for all carbons in viridiol (**10-C**). The largest deviations are bold in the table.

Atom	Exp.	M-I	M-II	M-IIIa	M-IIIb
1	60.7	60.8	57.5	59.8	59.5
2	81.7	82.0	78.9	94.5	90.2
3	61.6	68.5	64.7	63.8	63.8
4	122.1	121.0	125.9	130.9	128.5
**5**	145.6	146.3	144.5	155.5	153.2
6	173.4	178.9	173.0	153.8	154.8
**7**	142.4	140.4	146.6	149.6	149.4
8	42.3	45.7	39.2	51.9	49.0
9	71.7	73.7	85.9	81.7	76.2
10	30.5	20.2	23.5	31.5	32.4
**11**	158.6	153.9	149.3	**171.5**	**165.7**
12	127.3	127.3	125.3	137.4	132.1
13	127.3	127.4	128.4	133.3	132.2
14	129.8	133.2	134.1	141.4	143.9
**15**	158.0	**144.1**	**141.1**	165.0	168.8
16	136.9	126.5	126.4	141.2	137.7
**17**	145.7	148.0	145.3	**180.8**	**178.9**
18	28.4	28.4	23.5	32.9	33.2
19	36.5	36.2	36.2	38.9	39.6
20	206.7	206.3	206.2	214.2	211.7
RMSD		5.2	6.5	11.8	10.5

## Data Availability

All results shown in this article can be visualized by accessing the corresponding page on the WebCocon Server: https://cocon-nmr.de/publication_data (accessed on 8 January 2021).

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
