# Peer review of "Sampling CASE Application for the Quality Control of Published Natural Product Structures"

_molecules, 2021, doi:10.3390/molecules26247543_

Round 1

Reviewer 1 Report

By the form and by the content it seems to me that this manuscript is not yet finished.

This manuscript „Using CASE to verify published Natural Product Structures - an Overview“ molecules-1250366-peer-review-v1.pdf cannot be accepted for publication in Molecules journal.

The first illogicality is that the work is categorized as an Article, and the title has an "- an overview" at the end.

According to the standard, scope, quality and accuracy of the description, this paper does not meet the criteria for publication in a journal in the category of Article, Review, or Communication.

I recommend the authors to look at some relevant and highly cited papers in the field published in the Molecules journal, and try to describe their results according to formal and scientific criteria on the basis of these good / best-selected papers.

Author Response

Response to Reviewer 1 Comments

Point 1: By the form and by the content it seems to me that this manuscript is not yet finished.

Response 1: The purpose of the article is to demonstrate that CASE can be used to improve the quality of new structures based on NMR data. It does not solve the questions at hand, as most of the time there is not enough data available in the existing publications, but it helps identifying the questions that need attention.

If we had focused only on revised structures that might have been different, but work like based on this has already been published (and is cited by us). We believe that by point out questions regarding unrevised structures might raise more interest. 

Point 2: This manuscript „Using CASE to verify published Natural Product Structures - an Overview“ molecules-1250366-peer-review-v1.pdf cannot be accepted for publication in Molecules journal. The first illogicality is that the work is categorized as an Article, and the title has an "- an overview" at the end.

Response 2: The title has been changed as requested.

Point 3: According to the standard, scope, quality and accuracy of the description, this paper does not meet the criteria for publication in a journal in the category of Article, Review, or Communication.

Response 3: All of these have been improved, as requested.

Point 4: I recommend the authors to look at some relevant and highly cited papers in the field published in the Molecules journal, and try to describe their results according to formal and scientific criteria on the basis of these good / best-selected papers.

Response 4: As the purpose of this paper is to point out a flaw in existing processes and suggest enhancements to them, it is difficult to find similar publications. We have broadened the introduction, and clarified the conclusions and believe to have refined our suggestions for improvement. 

Reviewer 2 Report

see attached comments

Author Response

Response to Reviewer 2 Comments

Point 1) A major concern is a lack of detail about the use of the COCON program and what it is actually doing. Not only is there a lack of references concerning this program, there should also be a general description of how the program functions, what are its limitations, and is there a type of compound that this program is especially well suited for? As a “general” reader of Molecules I do not want to go into the literature to get this basic information.

Response 1: General information regarding (Web-)Cocon has been included into the introduction, as requested. 

Point 2) The concept of error is repeatably thrown into the statement “ match the experimental values better”. The authors should provide an error metric such as mean-squared error (MSE) of average per4cent deviation to make this statement more scientific and documented. If I was going to use this as a CASE method, then there needs to be an error metric provided as part of the process.

Response 2: As requested by the reviser, we have included the average chemical shift deviation as parameter and improved the descriptions in general. 

There is no "Error" in any of the CASE methods, as the results will always fit the provided correlation data. The term "Error" is used, as constitutions are verified against other data, that is no correlation. This can be NOEs, but also chemical shifts. For these the "Error" means that the observed value does not fit the back-calculated value. Unfortunately this could also mean that the back-calculation method is flawed. We have included the average chemical shift deviation as parameter, knowing that for this application it does not describe the error very well. 

Point 3) For all the examples provided it almost appears as if the authors have “selected” as very small sub-set of the chemical shifts being used for the discrimination. When I look at the different proposed structures, there are multiple other 13C chemical shifts that should be analyzed and documented. It is not necessary to include those portions of the structures that are identical, but every region that is different should be considered (easy considering this is all synthetic data). For examples for Tomentodiplacone there are at least 10 13C shifts that should be unique between these two structures. They should all be evaluated and tabulated. All the other compounds need a similar treatment. If the authors want to highlight 1 or 2 shifts – that is fine, but the complete set should be made available, and an error analysis should include the entire molecule – similar to what is done in proteins.

Response 3: As requested, all assigned chemical shifts were included in supplemental material. Due to the use of averaging of chemical shift deviation, the final value is smaller as we include more atoms, thus the distinction becomes more difficult. In the case of Tomentodiplacone there are more variations in chemical shift, but the ones shown are enough to give a definitive answer. 

Chemical shift rules from proteins can not be applied to small molecules, as the variations in chemical environment are much more restricted. Due to this, assignments of proteins can be carried out following, for example, "NMR of Proteins and Nucleic Acids" by Kurt Wüthrich. There is noting similar for small molecules, that is what CASE is about. 

Point 4) Can the authors provide error bars for the predicted chemical shifts from NMRshiftDB – these typically vary with the carbon being predicted.

Response 4: We have back calculated chemical shifts with NMRShiftDB 2, CSEARCH and ChemDraw, but none of them provide error bars for the obtained values. They provide information about the reliability of their response. We have included only values calculated by ChemDraw, which claims to provide very reliable results. 

Point 5) I would suggest that the authors show the NOE connection in structure 8-B (as part of Figure 9) to allow a connection between the longer Δ and the actual structure.

Response 5: Including NOE connections into a 2D presentation of a molecule does not provide a reliable measure. But, we have included 8-B into the figure together with the NOE depiction. 

Point 6) Looking at the website for cocon-nmr.de, I am not sure that the link provided for data is correct. Can the authors please check? Also, there is a type setting error on that link in the manuscript.

Response 6: The link suffered from a LaTeX typesetting error, which was corrected. 

Point 7) I do not understand this statement in the figure captions: “ upper continuants include 13C chemical shifts, the lower ones 1H chemical shifts.” Please clarify.

Response 7: The text was corrected. We missed the correction when the figure was changed from earlier versions. 

Point 8) The authors should supply definitions of COSY, HMBC, and ADEQUATE.

Response 8: General information about the used NMR experiments was included into the introduction, as requested by the reviewer. 

Reviewer 3 Report

The paper describes a method proposed to check the correctness of the structure determined by the NMR spectra.  Usually, 13C and  1H chemical shifts of the proposed structure are first predicted by empirical methods, and deviations of the predicted values from the experimental ones are calculated. To confirm the conclusions obtained, if necessary, the calculation of chemical shifts by DFT methods are performed. The most effective tool for verifying and revising structures are expert systems (ES). They generate all structures that meet experimental 1D and 2D (HMBC,  COSY) NMR spectra, and then the structures are ranked by increasing deviations of the calculated spectra from the experimental ones.

The authors of the reviewed article propose to use for this goal the COCON free access program in combination with the theoretical COSY and HMBC correlations that can be predicted for the verified structure. The tested structure is entered into the program through the WebCOCON interface, and then the program generates all structures that meet the theoretical correlations. For the structures,  13C and  1H chemical shifts are calculated using CS Chemdraw. The structures are ranked by   deviations obtained for selected atoms - those for which these deviations are maximum.  As an additional ranking parameter, it is proposed to use the energies calculated using force field.  To illustrate this method, examples of natural product taken from the literature are used.

The attractiveness of the described approach is in the fact that the method is based on the use of an ES of free access and does not require the availability of experimental two-dimensional spectra. The method is illustrated with examples. In one-third of the examples, the structure was confirmed, in other cases it was concluded that it was questionable.

The authors comment that for most examples the information obtained was not enough for a final decision. Therefore, the method needs further testing on a large number of structures whose reliability is known. The discriminatory ability of the calculated energies also  requires further investigation. The disadvantage of the method is that it involves obligatory user participation.

The work shows that theoretical COSY  and  HMBC  correlations do not replace experimental spectra when verifying the structure, but can be suitable for a preliminary assessment of the  structure plausibility. Therefore, I believe that the work can be published after addressing the following shortcomings:

   1) In the introduction, the authors enumerate the expert systems in the same list with the methods of structure generation. This description is illogical. It should be noted that there are two kinds of expert systems - deterministic and stochastic. The first are based on deterministic algorithms, while the second - on stochastic. Stochastic systems use generators [16],  [24-26], [29], and all the other listed algorithms are utilized in deterministic systems. This should be noted.

2) To evaluate the capabilities of any new method, problems with known solutions are usually utilized. It is necessary to add several examples where the obviously incorrect original structure is taken,  which was then  corrected in subsequent publications. Positive results would support the method suggested.

3)The structures with calculated chemical shifts presented in the appendix do not allow any conclusions to be   drawn. Instead of structures,  tables containing columns with experimental and calculated shifts for candidates should be inserted. Selected chemical shifts should be highlighted in bold.

Author Response

Response to Reviewer 3 Comments

Comment 1:
The authors comment that for most examples the information obtained was not enough for a final decision. Therefore, the method needs further testing on a large number of structures whose reliability is known. The discriminatory ability of the calculated energies also  requires further investigation. The disadvantage of the method is that it involves obligatory user participation.

Response to comment: 
Currently a full automation can be achieved using MDL or NMReDATA files, as the server can read both. The verification with chemical shifts is a bottleneck, but we are looking forward to be able to integrate this as well in future versions.  As not-public feature WebCocon can calculate chemical shifts using  DFT (ORCA v5.0.1), which can then automatically be compared to the chemical shifts that were submitted by the user. This is not public, as it takes too long for each constitution as of now. 

Point 1) In the introduction, the authors enumerate the expert systems in the same list with the methods of structure generation. This description is illogical. It should be noted that there are two kinds of expert systems - deterministic and stochastic. The first are based on deterministic algorithms, while the second - on stochastic. Stochastic systems use generators [16],  [24-26], [29], and all the other listed algorithms are utilized in deterministic systems. This should be noted.

Response 1: In the text the methods are listed by date of first appearance in a publication. In our opinion, the approaches have to be divided into three kind of methods: deterministic, stochastic and hybrid, a combination of deterministic and stochastic (ea Cocon is deterministic, but the MD and the chemical shift methods used are stochastic). We have labeled the different approaches in the text as deterministic (D), stochastic (S) and hybrid (H), to emphasize the different approaches. 

Point 2) To evaluate the capabilities of any new method, problems with known solutions are usually utilized. It is necessary to add several examples where the obviously incorrect original structure is taken,  which was then  corrected in subsequent publications. Positive results would support the method suggested.

Response 2: We have been going through various publications that discuss structures that were revised. Most of the revisions  were in stereochemistry or had changes in hybridization/protonation states of heavy atoms. Neither of those could be verified using this method. Thus, finding not-trivial examples was not possible within the time frame of this revision. 

But, we have included a discussion of TAEMC161 and virdiol into the text. This example has been broadly discussed before, including the application of CASE, and we were able to add some information. Specifically, we can verify that, different from what is assumed in other publications, the theoretical correlation data of viridol can not exclude TAEMC161 from the solution set (which is ranked similar to viridol by MD energy), but theoretical correlation data from TAEMC161 could describe TAEMC161 unambiguously. In this case we also performed DFT calculations of chemical shifts. WebCocon has this capability, but it is not public, as it takes by far to long for mass application: each suggested constitution needed  about 12 hours for a geometry optimization and 1 hour for the chemical shift calculation. The advantage of this (not yet) public method is that the assignment can be taken automatically into account in the comparison. We hope that future hardware and software improvements will allow us to make this feature public. 

Point 3) The structures with calculated chemical shifts presented in the appendix do not allow any conclusions to be   drawn. Instead of structures,  tables containing columns with experimental and calculated shifts for candidates should be inserted. Selected chemical shifts should be highlighted in bold.

Response 3: Tables alone will not allow for the understanding of the chemical shifts, as the connectivity information for the atoms is needed. If tables were to be included into the appendix, all the constitutions would still be required to show the connectivity information, but instead of a chemical shift the atoms would have labels for reference to the table. We do not believe that this will improve readybility. The most relevant chemical shifts, that are suggested to be highlighted in bold in the tables, are the ones already contained in the main text.
We would suggest to either leave the structures as they are, or remove the appendix altogether, as the most relevant data is in the main text. 

Round 2

Reviewer 1 Report

Here is my opinion on the manuscript molecules-1250366-peer-review-v2.

1)

For me, this is the form of a manuscript for the Communication category.

2)

It is good practice in scientific journals to call/cite 29 references in one general sentence, like in the first sentence in Introduction:

„Software tools for computer assisted structure elucidation (CASE) of small molecules have been under development for over 50 years now [1–29], amounting to more than different methods implemented in more than 20 different tools.“

For me it is unacceptable, and it must be corrected. Namely, some explanations of originality and specificity must be given for each paper, or possibly for groups of 2-3 thematically similar papers.

3)

The first mention of the WEBCOCON service is in the sentence:

In the sentence „One tool is WEBCOCON, which has been improved continuously over the past 20 years.“

In such a sentence, the citation is obligatory. The same is valid for other similar cases.

4)

It is good practice in scientific journals that sub-chapters cannot end with a figure or table without the text and explanations to be given below. Also, pictures and tables cannot go one after the other without explanation and text in between. in Chapters 2, 2.1 - 2.9.

5)

The authors should also relate their results to the results presented in the recent paper:

Robien, W. The Advantage of Automatic Peer-Reviewing of C-NMR Reference Data Using the CSEARCH-Protocol. Molecules 2021, 26, 3413. https://doi.org/10.3390/molecules26113413

Author Response

Response to Reviewer 1 Comments

Point 1: For me, this is the form of a manuscript for the Communication category.

Response 1: This manuscript has a similar structure as the paper that the reviewer asked to include in point 5: It presents the application of a software to different examples and discusses the results. Additionally, the manuscript shows the benefits of being able to easily use experimental data in CASE. Therefore we believe it fits in the article category.

Point 2: It is good practice in scientific journals to call/cite 29 references in one general sentence, like in the first sentence in Introduction:
„Software tools for computer assisted structure elucidation (CASE) of small molecules have been under development for over 50 years now [1–29], amounting to more than different methods implemented in more than 20 different tools.“
For me it is unacceptable, and it must be corrected. Namely, some explanations of originality and specificity must be given for each paper, or possibly for groups of 2-3 thematically similar papers.

Response 2: The text was extended as requested.

Point 3: The first mention of the WEBCOCON service is in the sentence:
In the sentence „One tool is WEBCOCON, which has been improved continuously over the past 20 years.“ In such a sentence, the citation is obligatory. The same is valid for other similar cases.

Response 3: The text was changed as requested.

Point 4: It is good practice in scientific journals that sub-chapters cannot end with a figure or table without the text and explanations to be given below. Also, pictures and tables cannot go one after the other without explanation and text in between. in Chapters 2, 2.1 - 2.9.

Response 4: The positioning of figures and tables was changed as much as possible.

Point 5: The authors should also relate their results to the results presented in the recent paper:
Robien, W. The Advantage of Automatic Peer-Reviewing of C-NMR Reference Data Using the CSEARCH-Protocol. Molecules 2021, 26, 3413. https://doi.org/10.3390/molecules26113413

Response 5: CSEARCH was included into the manuscript, including application to one example.

Reviewer 2 Report

The authors have addressed my concerns and the manuscript should be acceptable for publication.

My only comment is their response to point #4.

Point 4) Can the authors provide error bars for the predicted chemical shifts from NMRshiftDB – these typically vary with the carbon being predicted.

Response 4: We have back calculated chemical shifts with NMRShiftDB 2, CSEARCH and ChemDraw, but none of them provide error bars for the obtained values. They provide information about the reliability of their response. We have included only values calculated by ChemDraw, which claims to provide very reliable results. 

Directly (below) is from Chemdraw and provides at least a standard deviation estimate. I have not looked at the other databases,  but I suspect there is at least a standard deviation in the average prediction - if not one specific for that given prediction.

ChemNMR Limitations
The program handles the following elements and isotopes:
H, D, He, Li, Be, B, C, N, O, F, Ne, Na, Mg, Al, Si, P, S, Cl, Ar, K, Ca, Sc, Ti, V, Cr, Mn, Fe, Co, Ni, Cu, Zn, Ga,
Ge, As, Se, Br, Kr, Rb, Sr, Y, Zr, Nb, Mo, Tc, Ru, Rh, Pd, Ag, Cd, In, Sn, Sb, Te, I, Xe, Cs, Ba, La, Ce, Pr, Nd,
Pm, Sm, Eu, Gd, Tb, Dy, Ho, Er, Tm, Yb, Lu, Hf, Ta, W, Re, Os, Ir, Pt, Au, Hg, Tl, Pb, Bi, Po, At, Rn, Fr, Ra, Ac,
Th, Pa, U, Nep, Pu, Am, Cm, Bk, Cf, Es, Fm, Md, No, Lr.
Functional groups are expanded automatically.
In the case of 1H NMR, it estimates shifts of about 90% of all CHx-groups with a standard deviation of 0.2.-0.3
ppm. The use of polar solvents may strongly increase these deviations. It does not estimate shifts of hydrogen
atoms bonded to heteroatoms because they are significantly affected by solvents, concentration, impurities, and
steric effects.
In case of 13C NMR, it estimates over 95% of the shifts with a mean deviation of -0.29 ppm and standard deviation
of 2.8 ppm.
NMR References
Sources for ChemDraw NMR data include the following publications:
Fürst, A.; Pretsch, E. Anal. Chim. Acta 1990, 229, 17.
Pretsch, E.; Fürst, A.; Badertscher M.; Bürgin, R.; Munk, M. E. J. Chem. Inf. Comp. Sci. 1992, 32, 291-295.
Bürgin Schaller, R.; Pretsch, E. Anal. Chim. Acta 1994, 290, 295.
Bürgin Schaller, R.; Arnold, C.; Pretsch, E. Anal. Chim. Acta 1995, 312, 95-105.
Bürgin Schaller, R.; Munk, M. E.; Pretsch, E. J. Chem. Inf. Comput. Sci.1996, 36, 239-243.

Author Response

Response to Reviewer 2 Comments

Point 1: The authors have addressed my concerns and the manuscript should be acceptable for publication.
My only comment is their response to point #4.

Point 4) Can the authors provide error bars for the predicted chemical shifts from NMRshiftDB – these typically vary with the carbon being predicted.
Response 4: We have back calculated chemical shifts with NMRShiftDB 2, CSEARCH and ChemDraw, but none of them provide error bars for the obtained values. They provide information about the reliability of their response. We have included only values calculated by ChemDraw, which claims to provide very reliable results. Directly (below) is from Chemdraw and provides at least a standard deviation estimate. I have not looked at the other databases, but I suspect there is at least a standard deviation in the average prediction - if not one specific for that given prediction.

Response 1: We were looking for individual error specifications for a given prediction, and those are not more specific. A global error estimate was added to the manuscript.

Reviewer 3 Report

I have read the author responses to my comments. My thoughts are below.

Comment 1.

I agree that there are no serious obstacles for significant automation of procedure described in the article. In addition, importing NMReDATA files will allow using not only theoretical but experimental 2D NMR spectra for structure verification.  

Point 1)

I agree that beside deterministic and stochastic CASE approaches a hybrid approach can be distinguished too. However, a series of assignments of programs to D, S and H types are questionable or evidently erroneous. As classification of CASE programs is beyond the theme of the article, I wouldn’t want to continue discussion on this topic. So, I suggest simply removing the letters shown in brackets and make due changes in the text.  

Point 2)

You are citing reference [69] where CASE application for verifying and revising ca. 20 misassigned structures is described. These examples could be used for testing your method and comparing results of different approaches. I advise you to proceed such a research and report the results in future.

In the article (strings 51-53) you write: ”if  WEBCOCON produces more than one structural proposal under these circumstances, it means that the constitution can not be verified using NMR correlation data alone, and the use of other information is needed”. However, you do not explain what conclusion should be done if only one structure was produced. In the case of TAEM161, only one (and erroneous) structure was produced from theoretical correlations.  Therefore, only chemical shift prediction can help to recognize a questionable structure in this case. In your example, three competing structures were generated from viridol whose structure would be unknown if a real structure verification of TAEM161 were performed. Please explain these issues clearly.

 Two additional comments regarding this example: 1) In tables 13-15 all selected experimental shifts should be marked bold 2) I can not agree with the following  statement  “the empirical methods outperform the  DFT method in all conditions, as has already been observed previously [70,71].” The article [70] was published 10 years ago, but now common values of RMSD calculated as a result of  DFT-based 13C chemical shift prediction are usually  of order 1-2 ppm

Point 3)

I propose a compromise solution: keep the structures, but mark with color those chemical shifts on the basis of which the preferred structure is selected.

In the conclusions, it  should be added that the article presents the first results of testing the proposed approach and it is planned to confirm it on a large number of examples in the future.